# In-context Stochastic Gradient Descent with Hybrid Mamba-2 and Linear Self-Attention

## Abstract

State space models (SSMs) have recently emerged as a powerful alternative to Transformers by alleviating the quadratic computational overhead of self-attention. Among them, the Mamba-2 architecture (Dao & Gu, 2024) has been widely adopted in large language models. Despite this rapid progress, the theoretical foundations explaining how such models perform in-context learning (ICL) remain largely unclear. In this work, we provide a theoretical analysis of the Mamba-2 model and show that a single-layer Mamba-2 can simulate one step of gradient descent. Furthermore, we demonstrate that a hybrid architecture combining Mamba-2 with a Transformer—specifically, an SSD layer followed by a linear self-attention layer (SSD ∘ LSA)—can implement in-context stochastic gradient descent. Finally, we present experimental evidence that supports our theoretical results.

## 1 Introduction

Transformers (Vaswani et al., 2017) are highly capable models that have rapidly proliferated across real-world applications. A key contributor to their success is believed to be in-context learning (ICL) (Wei et al., 2023; Lu et al., 2023), a phenomenon first observed in natural language processing (NLP) tasks where large language models (LLMs) such as GPT-3 can make accurate predictions from only a few prompts without updating their parameters (Brown et al., 2020; Dong et al., 2022). However, the quadratic computational cost of Transformer self-attention has motivated the development of more efficient alternatives based on recurrent architectures. These include linear recurrent networks (Orvieto et al., 2023; Peng et al., 2024) and state space models (SSMs) (Gu & Dao, 2023; Dao & Gu, 2024), which reduce inference complexity to linear in sequence length. Recent advances in SSMs, such as Mamba, achieve performance competitive with Transformers at scale but still lag behind in ICL tasks (Park et al., 2024; Grazzi et al., 2024; Waleffe et al., 2024). Interestingly, hybrid models that combine Mamba with Transformers have been shown to exhibit ICL capabilities that can even surpass those of pure Transformers (Waleffe et al., 2024; Park et al., 2024). Yet, the mechanistic basis of how such hybrids perform ICL remains poorly understood. In this work, we take a constructive approach and demonstrate that the duality between the state space duality (SSD) layer in Mamba-2 (Dao & Gu, 2024) and linear self-attention (LSA) is central to enabling these hybrid models to achieve in-context learning.

A leading explanation for in-context learning (ICL) is that linear-attention Transformers implement gradient descent (GD) in-context (Von Oswald et al., 2023; Ahn et al., 2023). However, it remains unclear whether this perspective extends to hybrid architectures—particularly Mamba-2 models with SSD layers combined with Transformers, as employed in large-scale mixture-of-experts (MoE) systems (Team et al., 2025). To date, no explicit construction has been established for performing gradient descent on linear regression tasks with SSD ∘ LSA hybrid models; prior work has provided only empirical evidence (Grazzi et al., 2024; Waleffe et al., 2024; Park et al., 2024). This motivates our pursuit of a theoretical construction that can illuminate the mechanisms underlying the ICL capabilities of such hybrid models and offer practical explanatory insights.

In section 4, we explicitly construct SSD parameters by utilizing its duality with LSA to showcase that it can simulate one step of GD on linear regression tasks. We also conducted experiments to show that one layer SSD do mimic one step GD by plotting the test loss which shows great similarity

in pattern. We also show that the performance of multi-layer Mamba-2 doesn't improve as the layers get deeper, suggesting that multi-layer Mamba-2 may not perform multi-step GD.

In section 5, we first theoretically prove that a hybrid model SSD ∘ LSA, with a layer containing one SSD layer and one LSA layer, can implement multi-step SGD. The key insight is that the learnable "mask" of SSD controls which context example to attend to, and the value matrix of the LSA eliminates the change of $x$ while updates $y$ according to SGD. We then extend the result to show that one single hybrid layer can perform multi-step mini-batch SGD. We also conducted experiments on hybrid model with different layers to compare its performance with SGD and the original SSD models.

In summary, our contributions are to show that:

- We establish that a single layer of Mamba-2 is capable of simulating one step of gradient descent by utilizing the duality between SSD and LSA.

- By integrating SSD with LSA in a hybrid architecture, the model extends its capability from executing single-step updates to carrying out full multi-step stochastic gradient descent, effectively scaling from isolated updates to complete optimization procedures. We show that even a single hybrid layer suffices to implement multi-step SGD on a mini-batch.

- We empirically validate these theoretical findings: one-layer Mamba-2 closely mirrors the behavior of one-step gradient descent, while the composite architecture successfully emulates the iterative process of stochastic gradient descent.

## 2 RELATED WORK

**Transformer and ICL.** To better understand this capability, Garg et al. (2022) show that Transformers can perform in-context learning of various functions, including linear regression, two-layer neural networks, and decision trees. Theoretical studies of ICL have primarily focused on its connection to gradient descent. For example, Dai et al. (2022) identify a duality between Transformer attention and gradient descent, demonstrating that GPT-based ICL parallels explicit fine-tuning across multiple dimensions. Other works establish similar connections in simplified regression settings (Von Oswald et al., 2023; Ahn et al., 2023; Mavromatis et al., 2023; Li et al., 2024). In particular, Von Oswald et al. (2023) show that linear attention-only Transformers with carefully constructed parameters resemble models obtained by gradient descent, while Li et al. (2024) find analogous results for softmax attention-only Transformers. Beyond regression, recent studies explore richer function classes by analyzing self-attention with ReLU activations (Bai et al., 2023; Wang et al., 2024; Wu et al., 2024).

**SSM and ICL.** Since the introduction of the Mamba model by Gu & Dao (2023), its in-context learning (ICL) capabilities have attracted significant attention. Empirical studies (Park et al., 2024; Grazzi et al., 2024) have demonstrated Mamba's potential in ICL, with Grazzi et al. (2024) further proposing MambaFormer, a hybrid of Mamba and Transformer that achieves state-of-the-art ICL performance. Similarly, Waleffe et al. (2024) conduct systematic experiments on training Mamba models from scratch, showing that while Mamba underperforms Transformers in ICL, this limitation is alleviated when combined with Transformer layers. Indeed, large-scale commercial systems such as Team et al. (2025) have already adopted such hybrid architectures. From a theoretical standpoint, several works have explored the foundations of Mamba and related state space models (SSMs): Muca Cirone et al. (2024) link their expressiveness to linear controlled differential equations (CDEs); Halloran et al. (2024) show that Mamba's recurrent dynamics are robust to small perturbations in inputs; and Bondaschi et al. (2025) prove that even a single-layer Mamba can efficiently learn the in-context Laplacian smoothing estimator in Markov chain settings. More relevant to our work, Sushma et al. (2024) demonstrate that a structured SSM layer augmented with multiplicative input and output gating can replicate the behavior of an implicit linear model trained via one step of gradient descent with least squares loss. However, their analysis relies on a local sliding-window assumption that is not practical and does not address the less expressive state space duality (SSD) layer widely used in practice.

## 3 PRELIMINARIES

We define the input matrix to the sequence model as $Z = [z_1 \quad \cdots \quad z_{n+1}] \in \mathbb{R}^{D \times (n+1)}$, where each $z_i \in \mathbb{R}^D$ denotes a column of $Z$ corresponding to the $D$-dimensional embedding of a token.

### 3.1 IN-CONTEXT LEARNING

In the standard in-context learning (ICL) setting, a model is given a dataset $\mathcal{D} = \{(x_i, y_i)\}_{i \in [n]}$ along with a new test input $x_{n+1}$, where $\{x_i\}_{i=1}^n \subset \mathbb{R}^d$ are input vectors and $\{y_i\}_{i=1}^n$ are the corresponding labels. The inputs can be arranged into the matrix

$$Z = [z_1 \quad z_2 \quad \cdots \quad z_n \quad z_{n+1}] = \begin{bmatrix} x_1 & x_2 & \cdots & x_n & x_{n+1} \\ y_1 & y_2 & \cdots & y_n & 0 \end{bmatrix} \in \mathbb{R}^{(d+1) \times (n+1)}. \qquad (1)$$

Here the embedding dimension is $D = d + 1$, and the label corresponding to the test input is initialized to $0$. The model is then tasked with predicting $\hat{y}_{n+1}$, which should approximate the true label $y_{n+1}$ under a suitable evaluation metric, and filling it in the placeholder $0$ position. This theoretical formatting aligns with a line of previous works (Von Oswald et al., 2023; Ahn et al., 2023; Bai et al., 2023; Wang et al., 2024).

### 3.2 LINEAR SELF ATTENTION

A standard Transformer layer consists of both a self-attention mechanism and a feedforward MLP. In our setting, we focus on a simplified linear self-attention layer, which removes the original softmax activation. We further omit the MLP component, as it is not required in either our theoretical construction or our empirical validation.

> **Definition 3.1.** *(Linear self-attention) A linear self-attention (LSA) layer is denoted as* $\mathsf{LSA}_\theta(\cdot)$*, where* $\theta = \{W_v, W_q, W_k\} \subset \mathbb{R}^{D \times D}$*. The output of this layer on input* $Z \in \mathbb{R}^{D \times (n+1)}$ *is*
>
> $$\mathsf{LSA}_\theta(Z) = Z + \frac{1}{n} W_v Z (M \circ (Z^\top W_k^\top W_q Z)),$$
>
> *where* $M \in \mathbb{R}^{(n+1) \times (n+1)}$ *is the causal mask.*

The causal mask is a lower-triangular all-ones matrix. Throughout this work, we denote $P := W_v$ and $Q := W_k^\top W_q$, and define

$$\mathsf{LSA}_\theta(Z) = Z + \frac{1}{n} P Z (M \circ (Z^\top Q Z)). \qquad (2)$$

Prior studies (Von Oswald et al., 2023; Ahn et al., 2023; Gatmiry et al., 2024) have shown the effectiveness of linear self attention for enabling in-context learning. Furthermore, in our setting, Dao & Gu (2024) establish a state-space duality between Mamba-2 and linear self-attention, making the comparison between the two a natural choice.

### 3.3 MAMBA-2

A Mamba-2 model is a state space model (SSM) that maps the $t$-th token of the input matrix $z_t \in \mathbb{R}^D \mapsto y_t \in \mathbb{R}^D$ through an implicit latent state $H_t \in \mathbb{R}^{D \times N}$. Here $N$ is the kernel size of the latent state.

A general form of Mamba-2 take the form of

$$H_t = a_t \cdot H_{t-1} + z_t \otimes b_t \in \mathbb{R}^{D \times N},$$
$$y_t = H_t c_t \in \mathbb{R}^D,$$

where $a_t \in \mathbb{R}, b_t \in \mathbb{R}^N, c_t \in \mathbb{R}^N$ are learnable parameters.

By definition $H_1 = z_1 \otimes b_1$. Denote $a_{t:i} := a_t a_{t-1} \cdots a_{i+1}$ and $a_{t:t} := 1$. We have

$$
\begin{aligned}
H_t &= a_t \cdot H_{t-1} + Z_t \\
&= a_t a_{t-1} H_{t-2} + a_t Z_{t-1} + Z_t \\
&= \cdots \\
&= \sum_{s=1}^{t} a_{t:s} z_s \otimes b_s.
\end{aligned}
\tag{3}
$$

Now let's fix some notations. Let $B = [b_1, \cdots, b_{n+1}] \in \mathbb{R}^{N \times (n+1)}$, $C = [c_1, \cdots, c_{n+1}] \in \mathbb{R}^{N \times (n+1)}$. Suppose the parameters $a_t, b_t, c_t$ follows $[a_t, b_t, c_t] = \mathsf{Linear}(z_t)$ (Zhao et al., 2025), then we can denote $B = S_B Z$, $C = S_C Z$, where $S_B \in \mathbb{R}^{N \times N}, S_C \in \mathbb{R}^{N \times N}$ are parameter matrices. By rewriting eq. (3) in matrix form, we get the following form of Mamba-2.

---

**Definition 3.2.** *(Mamba-2) A Mamba-2 layer, also called the state space duality (SSD) layer, is denoted as $\mathsf{SSD}_\theta(\cdot)$, where $\theta = \{(L, S_B, S_C)\} \subset \mathbb{R}^{(n+1) \times (n+1)} \times \mathbb{R}^{D \times D} \times \mathbb{R}^{D \times D}$. The output of this layer on input $Z$ is*

$$
\mathsf{SSD}_\theta(Z) = Z + \frac{1}{n} Z(L \circ (Z^\top S_B^\top S_C Z)),
\tag{4}
$$

*where* $L^\top = \begin{bmatrix} 1 & & & & \\ a_2 & 1 & & & \\ a_{3:1} & a_3 & 1 & & \\ \vdots & \vdots & \ddots & \ddots & \\ a_{T:1} & a_{T:2} & \ldots & a_T & 1 \end{bmatrix}$ *is a learnable parameter matrix.*

---

Throughout this work we denote $S := S_B^\top S_C$ to consider

$$
\mathsf{SSD}_\theta(Z) = Z + \frac{1}{n} Z(L \circ (Z^\top S Z)).
\tag{5}
$$

Comparing the SSD layer (eq. (5)) with the LSA layer (eq. (2)), we observe a strong similarity. Although the SSD layer does not include the parameter matrix $P$, this role is effectively compensated by the structured parameter matrix $L$. Later, we will show that the matrices $P$ and $L$ play distinct roles in the in-context learning mechanisms of Mamba-2 and self-attention, respectively.

# 4 ONE LAYER MAMBA-2

We begin with the basic setting of a single-layer Mamba-2. Prior work on the in-context learning ability of linear self-attention (Von Oswald et al., 2023; Ahn et al., 2023) has shown that the selective accumulation of pairwise statistics allows the model to simulate gradient descent updates. The Mamba-2 architecture, as an SSM, also admits recurrence dynamics that can encode sufficient statistics of the training data. In section 4.1 we establish a parallel result: a single Mamba-2 layer can implement exactly one step of gradient descent (GD) when applied to an in-context linear regression task. This theoretical result is verified through experiments in section 4.2.

## 4.1 ONE LAYER MAMBA-2 PERFORMS ONE STEP GD

The ability to simulate gradient descent in-context is a fundamental property of architectures exhibiting algorithmic generalization. Linear self-attention (LSA) achieves this by exploiting inner-product structure to compute empirical gradients. Mamba-2, although not explicitly attention-based, is structurally similar: its selective state-space dynamics can recursively accumulate linear transformations of past tokens. This suggests that with a proper choice of parameters, the state updates of Mamba-2 can be engineered to compute gradient statistics and thereby simulate the GD update rule.

To theoretically validate the ability of Mamba-2, we aim to construct SSD parameters such that the input matrix forwarded once is equivalent to one-step of gradient descent.

**Theorem 4.1** (One layer Mamba-2 performs one step GD). *One layer Mamba-2 architecture can implement one-step gradient descent. Consider an in-context learning task with the input format in eq. (1). The data satisfy $y_i = w^\top x_i$ for $i \in [n]$. Let $y_{n+1}^1$ be the $(d+1, n+1)$-th output of the Mamba-2 layer, then there exists SSD parameters $\theta = \{(L, S_B, S_C)\}$ such that it holds that $y_{n+1}^1 = \langle w_1, x_{n+1} \rangle$, and the parameter $w_1$ follows the GD update:*

$$w_1 = w_0 - \eta \nabla L(w_0),$$

*where $w_0 = 0$ and $L(w) = \frac{1}{n} \sum_{i=1}^{n} (w^\top x_i - y_i)^2$.*

*Proof sketch.* We outline the main construction steps, deferring full details to section A.

By eq. (4), the output of a Mamba-2 layer at position $t$ can be expressed in the form

$$y_t = \sum_{i=1}^{t} a_{t:i} (u_i u_i^\top) S_B^\top S_C u_t,$$

where $u_t = [x_t; y_t]$ is the concatenation of feature and label at step $t$, $a_{t:i}$ are recurrence coefficients determined by the state-space recursion, and $S_B, S_C$ are learnable parameter matrices. To ensure that the update dynamics are driven only by the features and not by the labels of the current token, we choose $S_C$ so that $S_C u_t = [x_t; 0]$. In this way, only the feature vector passes forward to interact with the past states. $u_i u_i^\top$ expands as

$$\begin{bmatrix} x_i x_i^\top & x_i y_i \\ y_i x_i^\top & y_i^2 \end{bmatrix},$$

By appropriately designing the projection $S_B^\top$, the lower block of $(u_i u_i^\top) S_B^\top$ produces the vector $-\eta(w^\top x_i - y_i) x_i^\top$, which corresponds to the gradient of the squared loss on example $(x_i, y_i)$ scaled by a learning rate $\eta$. With initialization $w_0 = 0$, this term simplifies to $-\eta y_i x_i^\top$, making the updates purely linear in the data. As the layer processes the sequence of $n$ training examples, the hidden state accumulates these contributions to get

$$w_1 = \eta \sum_{i=1}^{n} y_i x_i,$$

which is exactly the result of applying one step of gradient descent from the origin on the linear regression objective. Finally, when the query token $x_{n+1}$ is presented, we apply a readout $h = [0_{P-1}; 1]$ that extracts the learned weight vector and outputs $w_1^\top x_{n+1}$, which corresponds precisely to the model's prediction after the single GD update. Thus, the construction shows how the Mamba-2 recurrence can be aligned with the algebraic structure of gradient descent. $\square$

**Remark 4.1.** *The result of theorem 4.1 can't generalize to multi-step gradient descent since after the first layer, the input $x$ would change, causing the next layer unable to utilize the original data. In an LSA multi-step GD is possible since the value matrix $P$ can ensure only the $y$ labels change during the forward. Thus we need more refined analysis to enable multi-step GD or similar iteration algorithms (see section 5.1).*

### 4.2 EXPERIMENTAL RESULTS FOR ONE LAYER MAMBA-2

For the linear regression dataset, we set the input dimension to $d = 10$ and the number of context examples during training to $n = 40$. Each feature vector is sampled as $x \sim \mathcal{N}(0, I_d)$, and the ground-truth weight vector is sampled as $w \sim \mathcal{N}(0, I_d)$. The labels are then generated according to the linear relation $y = \langle w, x \rangle$.

We trained several Mamba-2 models with varying depths ($l = 1, 4, 8, 12$ layers) using the ADAM optimizer with a fixed learning rate of $10^{-4}$. A total of 300,000 training samples were generated, organized into mini-batches of size 64. To reduce the risk of overfitting and to ensure that the models primarily capture in-context learning dynamics, we trained each model for only a single epoch.

For evaluation, we generated test datasets following the same distribution as in training but varied the number of in-context examples to assess generalization. Each reported performance is averaged over $k \in 10, 1000$ independent runs to reduce variance. As a baseline, we compared the models to standard gradient descent (GD) updates applied directly to the regression problem. Specifically, we first compared the performance of a one-layer Mamba-2 model with that of one-step GD, measured by squared loss across different context sizes ranging from 1 to 40.

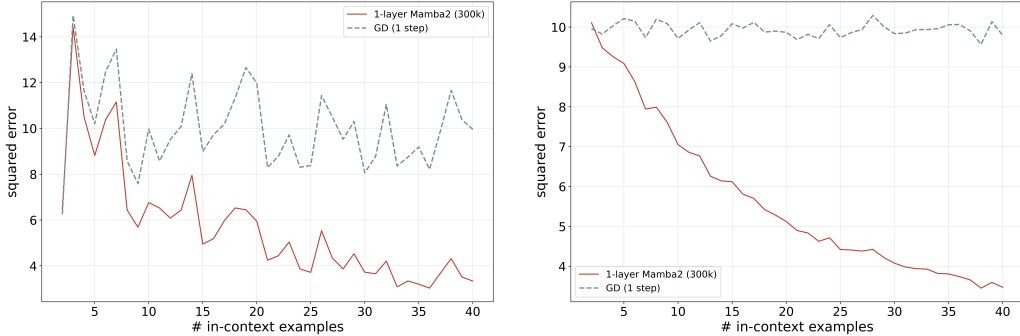

Figure 1: Comparison of one-layer Mamba-2 and one-step GD. *Left*: Each data point is averaged over 10 runs. *Right*: Each data point is averaged over 1000 runs.

The results are shown in fig. 1. On the left, we observe that the squared error for the one-layer Mamba-2 closely mirrors the behavior of one-step GD, following a nearly identical curve as the number of in-context examples increases. This suggests that a single Mamba-2 layer effectively implements a gradient-based update rule, akin to performing one step of GD, though with potentially different initializations or implicit parameterizations. On the right, we note that the Mamba-2 model consistently achieves lower squared error than one-step GD, despite following a similar trend.

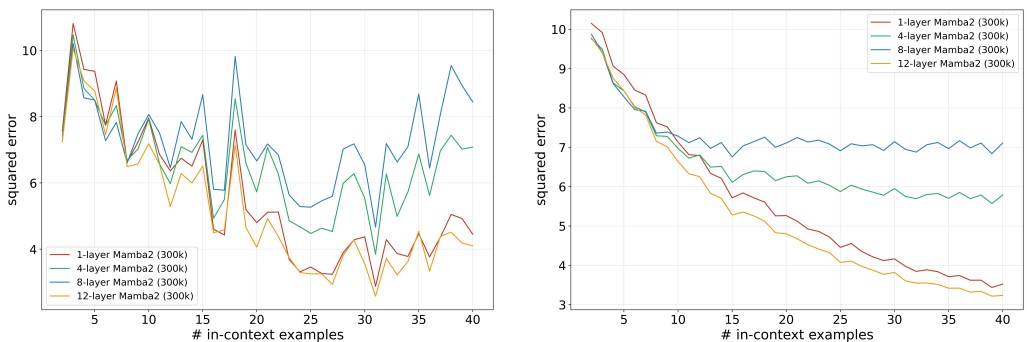

Figure 2: Comparison of Mamba-2 with different layers. *Left*: Each data point is averaged over 10 runs. *Right*: Each data point is averaged over 1000 runs.

Next, we examined the effect of increasing the number of layers in Mamba-2. As shown in fig. 2, increasing depth does not lead to monotonic improvements in performance. Interestingly, when the number of averaging runs $k$ is small, all models exhibit qualitatively similar performance patterns. This phenomenon may be attributed to the shared training schema, which constrains the models to learn similar update dynamics regardless of depth. These results suggest that simply stacking additional Mamba-2 layers does not straightforwardly enhance ICL performance in linear regression, and the benefit of depth may be limited without modifications to the model architecture itself.

## 5 HYBRID MAMBA PERFORMS SGD

Now we turn to the hybrid setting. Although one might hope that a sufficiently deep multi-layer Mamba-2 network could already realize multi-step gradient descent updates in-context, this is not

the case as discussed earlier. The fundamental reason is that the Mamba-2 block lacks a flexible value parameter matrix: it cannot directly perform the corrective update on the label $y$ that is essential for gradient-based adaptation while keeping the features $x$ unchanged. To overcome this, we consider a hybrid construction (which also follows practice) that alternates a Mamba-2 block with a linear self-attention block.

## 5.1 THE HYBRID ARCHITECTURE

We formalize our main theorem for the hybrid model below. The result shows that by carefully designing the interaction between the Mamba-2 and Transformer sublayers, an $l$-layer hybrid model can simulate $l$-step of stochastic gradient descent on linear regression tasks in the in-context setting.

---

**Theorem 5.1** (Hybrid model performs in-context SGD). *The SSD $\circ$ LSA hybrid architecture can implement in-context stochastic gradient descent. Consider an in-context learning task with the input format in eq. (1). The data satisfy $y_i = w^\top x_i$ for $i \in [n]$. Let $y_{n+1}^l$ be the $(d+1, n+1)$-th output of the $l$-th layer, then there exists proper parameters of the hybrid architecture such that $y_{n+1}^l = \langle w_l, x_{n+1} \rangle$, and $w_l$ follows the approximate SGD update:*

$$w_{l+1} = w_l - \eta \nabla L_t(w_l) + \epsilon_l,$$

*where $\|\epsilon_l\|_2 \le \epsilon$, $t = l \bmod n$, and $L_t(w) = \frac{1}{2}(w^\top x_t - y_t)^2$.*

---

*Proof sketch.* We give a high-level account of the main ideas and the full proof is deferred to section A.

Denote $l$ as the index of layer and $i$ as the index of token. Under the standing choices $A_l = -B_l$, $P_X^l = I_d$ and the uniform bounds $\|x\| \le C$, $\|y\| \le D$, the first SSD sublayer produces the update

$$x_i^{l+\frac{1}{2}} = x_i^l - \frac{1}{n} \sum_{j=1}^{n+1} L_{ji}^l \langle B_l x_i^l, x_j^l \rangle x_j^l,$$

from which one immediately obtains the residual bound $\|x_i^{l+\frac{1}{2}} - x_i^l\| \le \frac{1}{n} |\sum_j L_{ji}^l| \|B_l\| C^3$ showing that the per-sublayer change in $x$ is small: $\|x_i^{l+\frac{1}{2}} - x_i^l\| = \mathcal{O}(\|L^l\|_\infty \|B_l\|/n)$. Combining the SSD sublayer and the linear self-attention (LSA) sublayer into the net change $\Delta x_i = x_i^{l+1} - x_i^l$ we split $\Delta x_i$ into two parts $I_1$ and $I_2$, where $I_1$ captures the difference coming from using the slightly perturbed arguments $x^{l+\frac{1}{2}}$ versus $x^l$, and $I_2$ captures the deviation of the matrix weights from the identity. Using the pointwise expansion

$$\langle Bu, v \rangle w - \langle Bu_0, v_0 \rangle w_0 = \langle B(u - u_0), v \rangle w + \langle Bu_0, (v - v_0) \rangle w + \langle Bu_0, v_0 \rangle (w - w_0),$$

together with the smallness bound on $x^{l+\frac{1}{2}} - x^l$ yields $\|I_1\| = \mathcal{O}(\|B\|^2 \|L^l\|_\infty/n)$. By choosing $L^l$ so that only one column entry deviates from 1 (e.g. one entry equal to $1 - n\epsilon$) and by taking $\|B\| \le \epsilon$, the contribution $I_2$ can be made $\mathcal{O}(\epsilon^2)$; hence overall $\|\Delta x_i\| = \mathcal{O}(\epsilon^2)$ and the token features $x_i^l$ remain effectively constant across layers, justifying the replacement $x_i^l \approx x_i$. Turning to the scalar coordinates $y_i$, we write the two sublayer updates

$$y_i^{l+\frac{1}{2}} = y_i^l - \frac{1}{n} \sum_{j=1}^{n+1} L_{ji}^l \langle B_l x_i^l, x_j^l \rangle y_j^l \quad \text{and} \quad y_i^{l+1} = y_i^{l+\frac{1}{2}} - \frac{1}{n} \sum_{j=1}^{i} P_Y^l \langle A_l x_i^{l+\frac{1}{2}}, x_j^{l+\frac{1}{2}} \rangle y_j^{l+\frac{1}{2}},$$

and combine them to compute $y_i^{l+\frac{1}{2}} - y_i^l$ yields a leading term $-(L_{ti}^l - 1)n^{-1} \langle x_i, x_t \rangle y_t$ (with $t = l \bmod n$) plus error terms $E_1, E_2$. The term $E_1$, coming from the other (near-one) entries of $L^l$, is bounded by $\mathcal{O}(\epsilon/n)$, while $E_2$, which reflects the small differences between $x^{l+\frac{1}{2}}$ and $x^l$ and the corresponding $y$-increments, is likewise $\mathcal{O}(\epsilon/n)$ by the previous $\mathcal{O}(\|L^l\|_\infty \|B_l\|/n^2)$ estimate. Choosing $L_{ti}^l = 1 + \eta n$ makes the dominant contribution equal to $-\eta \langle x_i, x_t \rangle y_t$, so that up to the negligible $\mathcal{O}(\epsilon/n)$ errors we have

$$y_i^{l+1} - y_i^l = -\eta \langle x_i, x_t \rangle y_t + \mathcal{O}(\epsilon/n).$$

One then checks that this update structure preserves the affine form $y_i^l = y_i^0 + \langle \theta_l, x_i \rangle$ (with $\theta_l$ independent of $i$), and rewriting the main term reveals that the evolution of the layer parameter $\theta_l$ is equivalent to a gradient step on the squared-loss objective: letting $w_l = -\theta_l$ yields the claimed gradient-descent style update $\Delta y_i = \langle \eta \nabla L_t(-\theta_l), x_i \rangle + \mathcal{O}(\epsilon/n)$, which completes the sketch. $\qquad \square$

In theorem 5.1 we proved that one layer of hybrid model can perform one step of standard SGD, now we generalize the result to show that it can actually perform multi-step of mini-batch SGD.

**Theorem 5.2** (Hybrid model performs multi-step in-context SGD). *A single SSD $\circ$ LSA layer can implement multi-step in-context stochastic gradient descent. Consider an in-context learning task with the input format in eq. (1). The data satisfy $y_i = w^\top x_i$ for $i \in [n]$. Let $y_{n+1}^l$ be the $(d+1, n+1)$-th output of the l-th layer, and $S \subset [n]$ an index set of size $m$. Then there exists proper parameters of the hybrid architecture such that $y_{n+1}^l = \langle w_l^{(K)}, x_{n+1} \rangle$, and $w_l^{(K)}$ follows the approximate SGD update: for $r = 0, \cdots, K$:*

$$w^{(r+1)} = w^{(r)} - \eta \nabla L_S(w_l^{(r)}) + \epsilon_l^{(r)},$$

*where $\|\epsilon_l^{(r)}\|_2 \leq \epsilon$ and $L_S(w) = \frac{1}{m} \sum_{t \in S} (w^\top x_t - y_t)^2$.*

The proof idea is similar to theorem 5.1 thus we leave the details to section A.

**Discussion.** Theorem 5.1 establishes that the SSD $\circ$ LSA hybrid block can replicate the effect of a single SGD step per layer on linear regression data. Each layer consumes one training example $(x_t, y_t)$ and performs an approximate weight update $w_{l+1} = w_l - \eta \nabla L_t(w_l)$. The role of the Mamba sublayer is to accumulate the necessary inner products $\langle x_i, x_j \rangle$ and transmit them forward in the sequence, while the Transformer self-attention sublayer injects the value-update channel that modifies $y_i$ in accordance with the gradient. The error terms $E_1, E_2$ appearing in the proof are higher-order residuals that vanish as the sequence length $n$ grows. In theorem 5.2 we further generalize the result to show the ability of one layer hybrid model to perform multi-step mini-batch SGD. As far as we are concerned this is the first result to show that a single model layer can perform multi-step in-context iterative updates.

### 5.2 EXPERIMENTAL RESULTS FOR HYBRID MAMBA-2

The experimental setting is consistent with that described in section 4.2. In particular, the hybrid model adopts the SSD $\circ$ LSA stacking structure motivated by our theoretical construction. We trained hybrid models with varying numbers of layers ($l = 1, 4, 8, 12$) and evaluated their performance in comparison to stochastic gradient descent (SGD). The results are summarized in fig. 3 and fig. 4.

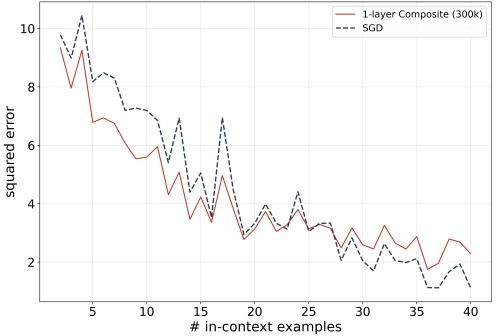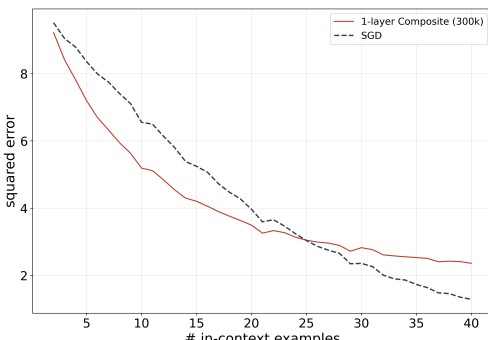

Figure 3: Comparison of SSD $\circ$ LSA hybrid model and SGD. *Left*: Each data point is averaged over 10 runs. *Right*: Each data point is averaged over 1000 runs.

On the left panel of fig. 3, where each data point is averaged over 10 independent runs, we observe that the one-layer hybrid model closely mimics the behavior of multi-step standard SGD across $n$

in-context points. This resemblance highlights the ability of a shallow hybrid model to approximate the dynamics of SGD in practice, collaborating our theorem 5.2. When the number of runs is significantly increased ($k = 1000$), as shown in the right panel, SGD slightly outperforms the one-layer hybrid model.

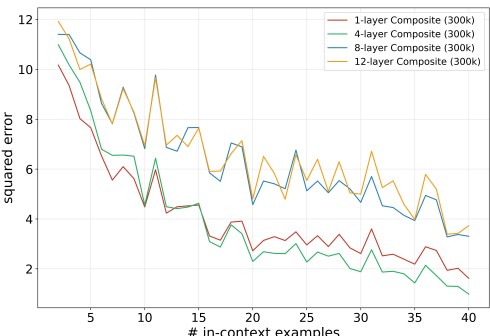 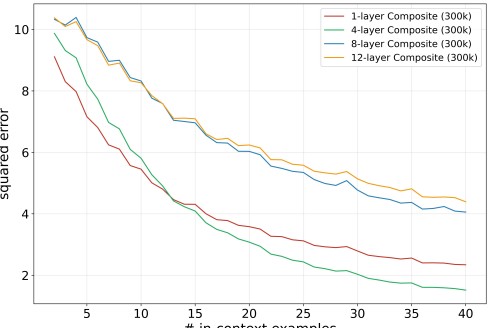

Figure 4: Comparison of SSD ∘ LSA hybrid model with different layers. *Left*: Each data point is averaged over 10 runs. *Right*: Each data point is averaged over 1000 runs.

We further investigated the effect of model depth on the performance of the hybrid architecture, with the results displayed in fig. 4. Again, the left panel shows outcomes averaged over 10 runs, while the right panel reports the results averaged over 1000 runs. A pattern consistent with the observations in section 4.2 emerges: when the averaging factor is small ($k = 10$), hybrid models with different depths exhibit nearly indistinguishable predictive behavior, and the variance across runs dominates the differences due to depth. When averaging over a larger number of runs ($k = 1000$), the performance differences become clearer, but deeper models do not always yield better results. In particular, while the 12-layer Mamba-2 model achieved the strongest performance in the pure SSD experiments, the best-performing hybrid model in this setting is the 4-layer variant. This finding suggests that in the SSD ∘ LSA hybrid architecture, there exists a trade-off between depth and effective approximation of the underlying SGD dynamics, and simply stacking more layers does not guarantee improved in-context learning performance.

## 6 CONCLUSION

In this work, we investigated the in-context learning (ICL) capabilities of Mamba-2 and its hybrid variants with Transformers through both theoretical construction and empirical validation. By leveraging the duality between the state space duality (SSD) layer in Mamba-2 and linear self-attention (LSA), we demonstrated that a single-layer Mamba-2 can simulate one step of gradient descent (GD) on linear regression tasks. This finding provides a principled explanation for previously observed empirical behaviors, where shallow Mamba-2 models mimic the dynamics of single-step optimization.

Building upon this foundation, we showed that hybrid architectures composed of SSD ∘ LSA layers can go beyond single-step updates to implement multi-step stochastic gradient descent (SGD). The key insight is that SSD selects which context examples contribute to the update, while LSA ensures the updates align with SGD dynamics by properly handling the interaction between inputs and outputs. Moreover, we proved that a single hybrid layer itself can implement multi-step SGD. Our experiments confirmed that hybrid models achieve multi-step optimization, thereby bridging the gap between empirical observations and theoretical understanding.

Overall, our results suggest that the combination of SSD and LSA provides a mechanistic foundation for the emergence of ICL in hybrid architectures. This perspective clarifies why Mamba-2 alone falls short of in-context learning, while SSD ∘ LSA hybrids can scale up to emulate full gradient-based learning procedures. We believe this work contributes a constructive theoretical framework for interpreting ICL in state space models and hybrid architectures used in practice, offering insights that may guide the design of more efficient large-scale models with enhanced in-context learning abilities.

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

# A  APPENDIX

Here we provide the proof for our main theorems.

**Theorem 4.1** (One layer Mamba-2 performs one step GD)**.** *One layer Mamba-2 architecture can implement one-step gradient descent. Consider an in-context learning task with the input format in eq. (1). The data satisfy $y_i = w^\top x_i$ for $i \in [n]$. Let $y_{n+1}^1$ be the $(d+1, n+1)$-th output of the Mamba-2 layer, then there exists SSD parameters $\theta = \{(L, S_B, S_C)\}$ such that it holds that $y_{n+1}^1 = \langle w_1, x_{n+1}\rangle$, and the parameter $w_1$ follows the GD update:*

$$w_1 = w_0 - \eta \nabla L(w_0),$$

*where $w_0 = 0$ and $L(w) = \frac{1}{n}\sum_{i=1}^{n}(w^\top x_i - y_i)^2$.*

*Proof.* Following the notations in . $X = [u_1, \cdots, u_T] \in \mathbb{R}^{P \times T}$, $B = [b_1, \cdots, b_T] \in \mathbb{R}^{N \times T}$, $C = [c_1, \cdots, c_T] \in \mathbb{R}^{N \times T}$, $a_{t:i} := a_t a_{t-1} \cdots a_{i+1}$ and $a_{t:t} := 1$.

Consider the Mamba-2 formula

$$Y = X(L \circ (B^\top C)),$$

where

$$L^\top = \begin{bmatrix} 1 & & & & \\ a_2 & 1 & & & \\ a_3 a_2 & a_3 & 1 & & \\ \vdots & \vdots & \ddots & \ddots & \\ a_\mathrm{T} \dots a_2 & a_\mathrm{T} \dots a_2 & \dots & a_\mathrm{T} & 1 \end{bmatrix}.$$

Then we have

$$y_t = \sum_{i=1}^{t} a_{t:i} u_i b_i^\top c_t$$

$$= \sum_{i=1}^{t} a_{t:i} Z_i c_t.$$

Now we define $C = S_C X$, $B = S_B X$. Then $c_t = S_C u_t$, $Z_i = u_i b_i^\top = u_i u_i^\top S_B^\top$. Thus

$$y_t = \sum_{i=1}^{t} a_{t:i} u_i u_i^\top S_B^\top S_C u_t.$$

For linear regression the loss function is

$$L(w) = \sum_{i=1}^{t} \|w^\top x_i - y_i\|^2,$$

and the gradient is

$$\nabla_w L(w) = \sum_{i=1}^{t} (w^\top x_i - y_i) x_i.$$

We hope that $u_t = [x_t; y_t]$ is updated to $u_t' = [x_t; y_t - \eta(\nabla_w L(w))^\top x_t]$. Now compare

$$\Delta u_t = \sum_{i=1}^{t} a_{t:i} u_i u_i^\top S_B^\top S_C u_t$$

and

$$\Delta y_t = -\eta \sum_{i=1}^{t} (w^\top x_i - y_i) x_i^\top x_t.$$

We hope for $i = 1, \cdots t$,

$$a_{t:i} u_i u_i^\top S_B^\top S_C u_t = [0_{P-1}; -\eta(w^\top x_i - y_i) x_i^\top x_t].$$

Let

$$S_C = \begin{bmatrix} I_{P-1} & 0_{P-1} \\ 0_{(N-P+1)\times(P-1)} & 0_{N-P+1} \end{bmatrix},$$

then $S_C u_t = [x_t; 0_{N-P+1}]$. Note that

$$u_i u_i^\top = \begin{bmatrix} x_i x_i^\top & x_i y_i \\ y_i x_i^\top & y_i^2 \end{bmatrix},$$

let's construct $S_B^\top$ such that

$$u_i u_i^\top S_B^\top = \begin{bmatrix} * & * \\ -\eta(w^\top x_i - y_i)x_i^\top & * \end{bmatrix}.$$

Initialize $w = 0$, then letting

$$S_B^\top = \begin{bmatrix} \eta I_{P-1} & * \\ * & * \end{bmatrix}$$

yields the desired result.

The output of the Mamba (a linear projection $h = [0_{P-1}; 1]$ applied to the last token) is

$$\eta \sum_{i=1}^{T} y_i x_i^\top x_T = w_1^\top x_T,$$

where $w_1$ is the output of one step of GD with learning rate $\eta$ and parameter initialization $w_0 = 0$. □

**Theorem 5.1** (Hybrid model performs in-context SGD). *The SSD ∘ LSA hybrid architecture can implement in-context stochastic gradient descent. Consider an in-context learning task with the input format in eq. (1). The data satisfy $y_i = w^\top x_i$ for $i \in [n]$. Let $y_{n+1}^l$ be the $(d+1, n+1)$-th output of the $l$-th layer, then there exists proper parameters of the hybrid architecture such that $y_{n+1}^l = \langle w_l, x_{n+1} \rangle$, and $w_l$ follows the approximate SGD update:*

$$w_{l+1} = w_l - \eta \nabla L_t(w_l) + \epsilon_l,$$

*where $\|\epsilon_l\|_2 \leq \epsilon$, $t = l \bmod n$, and $L_t(w) = \frac{1}{2}(w^\top x_t - y_t)^2$.*

*Proof.* We set $A_l = -B_l$ and $P_X^l = I_d$ and abbreviate $\| \cdot \|_2$ as $\| \cdot \|$. Suppose $\|x\| \leq C$ and $\|y\| \leq D$.

For the first SSD sublayer the update formula is:

$$x_i^{l+\frac{1}{2}} = x_i^l - \frac{1}{n} \sum_{j=1}^{n+1} L_{ji}^l \langle B_l x_i^l, x_j^l \rangle x_j^l.$$

The residual can be bounded by

$$\|x_i^{l+\frac{1}{2}} - x_i^l\| \leq \frac{1}{n}|\sum_j L_{ji}^l|\|B_l\|C^3 = \mathcal{O}\left(\frac{\|L^l\|_\infty \|B_l\|}{n}\right), \tag{6}$$

where $\|L^l\|_\infty = \max_i |\sum_j L_{ji}^l|$.

For the second linear self attention sublayer the update formula is:

$$x_i^{l+1} = x_i^{l+\frac{1}{2}} - \frac{1}{n} \sum_{j=1}^{n+1} \langle -B_l x_i^{l+\frac{1}{2}}, x_j^{l+\frac{1}{2}} \rangle x_j^{l+\frac{1}{2}}.$$

Combine the two updates into net change from $x_i^l$ to $x_i^{l+1}$:

$$x_i^{l+1} - x_i^l = -\frac{1}{n} \sum_j L_{ji}^l \langle B_l x_i^l, x_j^l \rangle x_j^l - \frac{1}{n} \sum_j \langle -B_l x_i^{l+\frac{1}{2}}, x_j^{l+\frac{1}{2}} \rangle x_j^{l+\frac{1}{2}}.$$

Then

$$
\begin{aligned}
\Delta x_i &= x_i^{l+1} - x_i^l \\
&= \underbrace{\frac{1}{n}\sum_j (\langle B_l x_i^{l+\frac{1}{2}}, x_j^{l+\frac{1}{2}}\rangle x_j^{l+\frac{1}{2}} - \langle B_l x_i^l, x_j^l\rangle x_j^l)}_{I_1} + \underbrace{\frac{1}{n}\sum_j (1 - L_{ji}^l)\langle B_l x_i^l, x_j^l\rangle x_j^l}_{I_2}.
\end{aligned}
$$

To bound $I_1$, note that

$$
\langle Bu, v\rangle w - \langle Bu_0, v_0\rangle w_0 = \langle B(u - v_0), v\rangle w + \langle Bu_0, (v - v_0)\rangle w + \langle Bu_0, v_0\rangle (w - w_0),
$$

and plug in $u = x_i^{l+\frac{1}{2}}, u_0 = x_i^l, v_0 = x_j^l, w = x_j^{l+\frac{1}{2}}, w_0 = x_j^l$. Using $\|x\| \le C$ and $\|B_l\| = \|B\|$ for short and $\|x_i^{l+\frac{1}{2}} - x_i^l\| \le \delta$ (eq. (6)), each of the three terms is bounded by $\|B\|C^2\delta$. Thus

$$
\|I_1\| \le \frac{1}{n}\sum_j 3\|B\|C^2\delta \le 3\|B\|C^2\delta = \mathcal{O}\left(\frac{\|B\|^2\|L^l\|_\infty}{n}\right).
$$

To bound $I_2$ we directly see

$$
\|I_2\| \le \frac{1}{n}\sum_j |1 - L_{ji}^l||\langle B_l x_i^l, x_j^l\rangle|\|x_j^l\| \le \frac{\|B\|C^3}{n}\sum_j |1 - L_{ji}^l|.
$$

By choosing proper $L_{ji}^l$ (i.e. only one non-one element $1 - n\epsilon$ for fixed $i$) we can ensure $\sum_j |1 - L_{ji}^l| \le n\epsilon$. Note that by this choice $\|L^l\|_\infty = \mathcal{O}(n)$. Also we can always ensure $\|B\| \le \epsilon$. Thus

$$
\|\Delta x_i\| = \mathcal{O}(\epsilon^2).
$$

Since the change of $x$ is small, we can denote $x_i^l$ as $x_i$ because they're almost the same for each $l$.

Now let's consider the update of $y$. We know that for the first sublayer update:

$$
y_i^{l+\frac{1}{2}} = y_i^l - \frac{1}{n}\sum_{j=1}^{n+1} L_{ji}^l\langle B_l x_i^l, x_j^l\rangle y_j^l, \tag{7}
$$

and for the second sublayer update:

$$
y_i^{l+1} = y_i^{l+\frac{1}{2}} - \frac{1}{n}\sum_{j=1}^{i} P_Y^l\langle A_l x_i^{l+\frac{1}{2}}, x_j^{l+\frac{1}{2}}\rangle y_j^{l+\frac{1}{2}}. \tag{8}
$$

Thus if we set $P_Y^l = -1$ and denote $t := l\%n$, then

$$
\begin{aligned}
\Delta y_i &= y_i^{l+1} - y_i^l \\
&= -\frac{1}{n}\sum_{j=1}^{i}(L_{ji}^l\langle B_l x_i^l, x_j^l\rangle y_j^l + P_Y^l\langle A_l x_i^{l+\frac{1}{2}}, x_j^{l+\frac{1}{2}}\rangle y_j^{l+\frac{1}{2}}) \\
&= -\frac{1}{n}\sum_{j=1}^{i}(L_{ji}^l\langle B_l x_i^l, x_j^l\rangle y_j^l + \langle B_l x_i^{l+\frac{1}{2}}, x_j^{l+\frac{1}{2}}\rangle y_j^{l+\frac{1}{2}}) \\
&= -\frac{L_{ti} - 1}{n}\langle x_i^l, x_t^l\rangle y_t^l - \underbrace{\frac{1}{n}\sum_{j\ne t}(L_{ji} - 1)\langle x_i^l, x_j^l\rangle y_j^l}_{E_1} - \underbrace{\frac{1}{n}\sum_{j=1}^{i}(\langle x_i^{l+\frac{1}{2}}, x_j^{l+\frac{1}{2}}\rangle y_j^{l+\frac{1}{2}} - \langle x_i^l, x_j^l\rangle y_j^l)}_{E_2}.
\end{aligned}
$$

Now let's bound $E_1$ and $E_2$ separately.

$$
|E_1| \le \frac{1}{n}|\sum_{j\ne t}(L_{ji}^l - 1)|\langle x_i^l, x_j^l\rangle| \cdot |y_j^l|| \le \frac{C^2 D}{n}|\sum_{j\ne t}(L_{ji}^l - 1)| = \mathcal{O}\left(\frac{\epsilon}{n}\right).
$$

Since

$$\langle x_i^{l+\frac{1}{2}}, x_j^{l+\frac{1}{2}}\rangle y_j^{l+\frac{1}{2}} - \langle x_i^l, x_j^l\rangle y_j^l = \langle x_i^{l+\frac{1}{2}} - x_i^l, x_j^l\rangle y_j^l + \langle x_i^l, x_j^{l+\frac{1}{2}} - x_j^l\rangle y_j^l$$
$$+ \langle x_i^{l+\frac{1}{2}}, x_j^{l+\frac{1}{2}}\rangle (y_j^{l+\frac{1}{2}} - y_j^l),$$

by triangle inequality there exists a constant $K$ (depending only on $C$ and finite-dim constants) such that

$$|E_2| \leq \frac{K}{n}\sum_{j=1}^{i}\left(\|x_i^{l+\frac{1}{2}} - x_i^l\| + \|x_j^{l+\frac{1}{2}} - x_j^l\| + |y_j^{l+\frac{1}{2}} - y_j^l|\right) = \mathcal{O}\left(\frac{\|L^l\|_\infty \|B_l\|}{n^2}\right) = \mathcal{O}\left(\frac{\epsilon}{n}\right).$$

Choose $L_{ti}^l = 1 + \eta n$ and by the update formula of $y$ (eq. (7) and eq. (8)) we can verify that $y_i^l = y_i^0 + \langle \theta_l, x_i\rangle$ (i.e. $\theta_l$ doesn't depend on index $i$), so the update for $y$ follows

$$
\begin{aligned}
y_i^{l+1} - y_i^l &= -\eta\langle x_i, x_t\rangle y_t^l + \mathcal{O}(\epsilon) \\
&= -\eta\langle y_t^l x_t^\top, x_i\rangle + \mathcal{O}(\epsilon) \\
&= -\langle \eta(y_t^0 + \theta_l^\top x_t)x_t^\top, x_i\rangle + \mathcal{O}(\epsilon) \\
&= \langle \eta\nabla L_t(-\theta_l), x_i\rangle + \mathcal{O}(\epsilon),
\end{aligned}
$$

then letting $w_l = -\theta_l$ and we get the desired result. $\qquad\square$

**Theorem 5.2** (Hybrid model performs multi-step in-context SGD). *A single SSD $\circ$ LSA layer can implement multi-step in-context stochastic gradient descent. Consider an in-context learning task with the input format in eq. (1). The data satisfy $y_i = w^\top x_i$ for $i \in [n]$. Let $y_{n+1}^l$ be the $(d+1, n+1)$-th output of the $l$-th layer, and $S \subset [n]$ an index set of size $m$. Then there exists proper parameters of the hybrid architecture such that $y_{n+1}^l = \langle w_l^{(K)}, x_{n+1}\rangle$, and $w_l^{(K)}$ follows the approximate SGD update: for $r = 0,\cdots,K$:*

$$w^{(r+1)} = w^{(r)} - \eta\nabla L_S(w_l^{(r)}) + \epsilon_l^{(r)},$$

*where $\|\epsilon_l^{(r)}\|_2 \leq \epsilon$ and $L_S(w) = \frac{1}{m}\sum_{t \in S}(w^\top x_t - y_t)^2$.*

*Proof.* We first define

$$R := \sum_{t \in S} x_t x_t^\top \in \mathbb{R}^{d \times d}.$$

Define a linear operator $T : \mathbb{R}^n \to \mathbb{R}^n$ which acts on vector $y \in \mathbb{R}^n$ by

$$(Ty)_i = \frac{1}{m}\sum_{t \in S}\langle x_i, x_t\rangle y_t, \quad i = 1,\cdots,n.$$

By the proof of theorem 5.1 we know that a single-step SGD operator acts on the $y$ vector by the linear operator

$$M = I - \eta T.$$

Hence $K$ steps of SGD correspond to applying $M$ to $y$ for $K$ times:

$$y^{(K)} = M^K y^{(0)} = (1 - \eta T)^K y^{(0)}.$$

Define the increment operator

$$\Delta M := M^K - I = (1 - \eta T)^K - I. \tag{9}$$

By the binomial expansion

$$\Delta M = \sum_{s=1}^{K}\binom{K}{s}(-\eta)^s T^s.$$

We observe that $T^s$ has an entrywise representation built from matrix $R$:

$$(T^s)_{ij} = \frac{1}{m} x_i^\top R^{s-1} x_j \quad \text{for all } i, j, \tag{10}$$

and moreover $T_{ij}^s = 0$ when $j \notin S$. (We can prove eq. (10) by induction.)

By eq. (9) and eq. (10) we can collect the polynomial in powers of $S$ and write:

$$\Delta M_{ij} = 1_{j \in S} \cdot x_i^\top Q x_j, \quad \text{where} \quad Q = \sum_{s=1}^{K} \binom{K}{s} (-\eta)^s \frac{1}{m^s} S^{s-1}. \tag{11}$$

So $\Delta M$ has a column-sparse structure: only columns indexed by $j \in S$ are nonzero. The $K$-step SGD we want the hybrid layer to implement on $y$ is:

$$y^{(K)} = y^{(0)} + \Delta M y^{(0)}.$$

By eq. (7) and eq. (8) we know that the increment of $y_i$ is

$$\Delta y_i = -\frac{1}{n} \sum_{j=1}^{N} (L_{ji} \langle Bx_i, x_j \rangle y_j + \langle -Bx_i^{\frac{1}{2}}, x_j^{\frac{1}{2}} \rangle y_j^{\frac{1}{2}}),$$

where we omit the layer index $l$. And by the same argument in the proof for theorem 5.1 we obtain the leading-term approximation:

$$\Delta y_i = -\frac{1}{n} \sum_{j=1}^{N} (L_{ji} - 1) \langle Bx_i, x_j \rangle y_j + E. \tag{12}$$

We set $B = Q$ and $L_{ji} = 1 - n \cdot 1_{j \in S}$, then the leading term in eq. (12) becomes

$$-\frac{1}{n} \sum_{j} (L_{ji} - 1) \langle Bx_i, x_j \rangle y_j = -\frac{1}{n} \sum_{j \in S} (-n) x_i^\top Q x_j y_j = (\Delta M y)_i,$$

so the SGD is recovered if no residual exists.

Now let's turn to the residuals in $E$. With the same argument in the previous proof (the $I_1, I_2$ decomposition) for $\Delta x_i = x_i^{\frac{1}{2}} - x_i$ we obtain $\|\Delta x_i\| = \mathcal{O}(m\epsilon)$.

For residuals in the $y$ update we also adopt the $E_1, E_2$ decomposition. $E_1$ denotes contributions from indices not in $S$. By our choice of $L$ we know

$$|E_1| = 0.$$

For $E_2$ we have

$$|E_2| \leq \frac{K_0}{n} \sum_{j=1}^{i} (\|x_i^{\frac{1}{2}} - x_i\| + \|x_j^{\frac{1}{2}} - x_j\| + |y_j^{\frac{1}{2}} - y_j|) = \frac{K_0 i}{n} (2\delta_x + Cm\|B\|) = \mathcal{O}(m\|B\|).$$

Since we set $Q = B$, we need to ensure $Q$ is bounded by stepsize $\eta$. We know that

$$\|Q\| = \frac{1}{m} \sum_{s=1}^{K} \binom{K}{s} |\eta|^s C^{2(s-1)} = \frac{|\eta|}{m} \sum_{s=1}^{K} \binom{K}{s} (|\eta| C^2)^{s-1}.$$

Since

$$\sum_{s=1}^{K} \binom{K}{s} a^{s-1} \leq \frac{(1+a)^K - 1}{a},$$

we set $a = |\eta| C^2$ and

$$\|Q\| \leq \frac{1}{mC^2} ((1 + |\eta| C^2)^K - 1).$$

For small $|\eta| C^2$ we can further utilize $(1 + |\eta| C^2)^K - 1 \approx K|\eta| C^2$, giving $\|Q\| = \mathcal{O}(K/m)|\eta|$, so $\|Q\|$ is indeed restricted by a small stepsize $\eta$. Thus setting $\epsilon$ and $\eta$ at the same order would guarantee that the residual is of order $|E| = \mathcal{O}(\epsilon)$. This gives the desired result. $\qquad \square$

