# OpenReview forum: "In-Context Stochastic Gradient Descent with Hybrid Mamba-2 and Linear Self-Attention Model"
_ICLR.cc/2026/Conference — Submitted to ICLR 2026_

### Official Review · Reviewer_mZkm · 2025-10-20

**Soundness:** 2
**Presentation:** 2
**Contribution:** 2
**Rating:** 2
**Confidence:** 3

**Summary:**

This paper theoretically proves that:
- single-layer Mamba-2 can simulate one step of gradient descent for in-context learning (ICL).
- hybrid Mamba-2 and linear self-attention(LSA) can simulate stochastic gradient descent for ICL.
- The error of simulating SGD is well bounded, and the theoretical results are verified by experiments.

**Strengths:**

1. This paper presents an interesting theoretical result on ICL for hybrid Mamba-2 and LSA. i.e., simulating SGD for ICL.
2. This paper provides experimental results to support the theoretical results.
3. Paper is well structured.

**Weaknesses:**

1. The claimed connection between SSMs and SGD is not particularly novel, and similar ideas have been discussed in previous works.
- Yang, Songlin, et al. Gated Delta Networks: Improving Mamba2 with Delta Rule. At ICLR 2025.
- Behrouz, Ali, et al. Atlas: Learning to optimally memorize the context at test time. arXiv preprint arXiv:2505.23735 (2025).
- Jiang, Jiarui, et al. Trained Mamba Emulates Online Gradient Descent in In-Context Linear Regression. At NeurIPS 2025.
- Li, Yingcong, et al. Gating is weighting: Understanding gated linear attention through in-context learning. At COLM 2025

Thus, the **novelty should be better clarified**.

2. The assumption seems too strong: The SGD pattern relies on the construction of the model's weights, and this paper focuses on how these weights work (expresivity), but it does not guarantee that the model can be trained to such weights (trainability). The difficulty of proof is not high; therefore, there is a **lack of technical contribution**.

3. Some experiments are redundant: For example, 10 runs vs. 1000 runs, they share similar meanings. I also recommend that the authors plot the error bars in the experiments.

Minors and Typos:

I suggest that vectors and matrices should be **bolded**.

**Questions:**

1. Is the experiments follow the theoretical construction? I think it should be clarified in the paper.

2. This paper theoretically claims that hybrid model simulates SGD for ICL, but the experimental results only compare the squared error between models and (S)GD. Comparing their error is not enough to show that hybrid model actually simulates SGD. I recommend the authors to present more evidence to show that.

---

### Official Review · Reviewer_AXmp · 2025-10-21

**Soundness:** 3
**Presentation:** 2
**Contribution:** 2
**Rating:** 6
**Confidence:** 2

**Summary:**

The paper shows that a model with linear state-space layers (SSD) and linear self-attention layers (LSA) can perform ICL via gradient descent. The property follows from the structure of the cubic transformations associated with SSD and LSA layers.

**Strengths:**

- Linear and hybrid state-space models are still outperformed by transformers on several tasks. The work may clarify the role of nonlinearities in ICL and help find better complexity-performance tradeoffs.
- Remark 4.1 seems to explain why linear state-space models do not perform well on ICL tasks.

**Weaknesses:**

- The paper's main motivation is understanding whether ICL extends to hybrid architectures. As hybrid architectures include linear ones as a special case, the authors should clarify how this may *not* happen.
- SSD layers are equivalent to linear state-space models. The authors show that they are connected to self-attention layers by a rotation of the input. They should clarify why they expect advantages by alternating between SSD and self-attention layers.
- The authors should also explain in theory and show empirically why their construction is expected to outperform models with LSA layers only.
- The main theoretical conclusion (according to the paper's motivation stated in the abstract) is in Remark 4.1. The authors should formalise it and give more details.

**Questions:**

- Can you clarify the following sentence, * ''the duality between the state space duality (SSD) layer in Mamba-2 and linear self-attention is central to enabling these hybrid models to achieve in-context learning.'' *
- The word *duality* is used several times without a clear definition. Is it used as a synonym of equivalence?
- Is the formulation of LSA as in Equation 2 new?
- Can the difference between SSD and LSA be viewed as a rotation of the input?
- I do not fully understand the meaning of *selective* and *simulate* in * ''the selective accumulation of pairwise statistics allows the model to simulate gradient descent updates'' *.
- Does Theorem 4.1 hold even if $w_0 \neq 0$?
- How do you choose $\eta$ in one-step GD?
- I do not understand why you need two plots in the figures.
- Theorem 5.1 suggests that SSD layers add noise to the clean GD steps performed by LSA layers. Is this correct? Why can they be expected to improve the model performance then?

---

### Official Review · Reviewer_eSv3 · 2025-11-03

**Soundness:** 1
**Presentation:** 2
**Contribution:** 1
**Rating:** 2
**Confidence:** 4

**Summary:**

This paper studies the in-context learning of linear regressions with Mamba-2 and a hybrid model. The authors begin by showing the existence of a one-layer Mamba-2 model that is simulating one-step of gradient descent (GD) in its forward pass. The authors further compare the errors of Mamba-2 models and one-step GD in experiments. Then they study the hybrid layer of a Mamba-2 layer followed by a linear self-attention layer, claiming that a single hybrid layer could implement multiple steps of stochastic gradient descent (SGD) in-context. Finally the authors evalute hybrid models of different depths and compare them with SGD.

**Strengths:**

The authors uses the duality between Mamba-2 and causal linear self-attention. Hence the construction of one-layer Mamba-2 performing GD in-context is convincing.

**Weaknesses:**

1. The section (Sec 5.1) of hybrid model is confusing and questionable. The first theorem of Sec 5.1 claims a one-layer hybrid model could approximate one step of SGD. The statement of the theorem seems inaccurate and the proof appears to be unreliable to me. The authors are taking the length of context to infinity but this is not mentioned in the theorem. The proof relies on that the learned weights matrix (B) is small, which does not seem to be true by the construction. The second theorem surprisingly states that a single hybrid layer could perform multiple steps of SGD. However, from the proof, the constructed weights matrix seems to depend on the input context, which makes the theorem incorrect.

2. The experiments do not really justify the theory and some hyperparameters are missing. In Fig 1, the performance of one-layer Mamba is much better than one step of GD, and the learning rate of GD is not reported. The error curve of GD is a bit confusing since the error does not decrease as the number of in-context example increases.

See questions for details.

**Questions:**

1. In the proof to Theorem 5.1, at line 364 why is $\|B\|\leq \epsilon$? At line 749, the constructed $B$ seems to be identity.
2. Line 372, is there a specific reason of choosing $t=l \text{ mod } n$?
3. In the proof to Theorem 5.2, by definition of operator $T$, the increment $\Delta M$ in eq(9) should be a degree-$(2K)$ function of $x$ because $T$ is in degree-$2$ of $x$. Here $K$ is the number of GD. Then by equation (11) $Q$ should be in degree-$(2K-2)$ of $x$, and we have the weight matrix $B=Q$. Does this suggest that for this specific construction, any $K>1$ should cause the weight matrix depending on the input context?
4. What is the learning rate for GD in Fig 1? Why does the error curve of GD keep flat while increasing the number of in-context examples?
5. What is the number of steps of SGD in Fig 3?

---

### Meta-Review · Area_Chair_s7Lb · 2026-01-01

**Summary:**

Consensus by reviewers to reject and no author rebuttal.

**Reviewer Concerns:**

N/A - no author rebuttal.

**Reviewer Scores:**

N/A - no author rebuttal.

---

### Decision · Program_Chairs · 2026-01-26

Reject